



# Experimental Investigation of Aerodynamic Characteristics of Bat Carcasses after Collision with a Wind Turbine

Shivendra Prakash[1,2] and Corey D. Markfort[1,2]

[1]IIHR-Hydroscience and Engineering, The University of Iowa, Iowa City, IA 52242, USA.
[2]Civil and Environmental Engineering, The University of Iowa, Iowa City, IA 52242, USA.

*Correspondence to*: Corey D. Markfort (corey-markfort@uiowa.edu)

**Abstract.** Large number of bat fatalities have been reported in wind energy facilities in different parts of the world. The wind farm regulators are required to monitor the bat fatalities by conducting carcass survey in the wind farms. Previous
studies have implemented ballistic model to characterize the carcass fall zone after strike with turbine blades. Ballistic model contains the aerodynamic drag force term which is dependent upon carcass drag coefficient. The bat carcass drag coefficient is highly uncertain and of which no measurement is available. This manuscript introduces a new methodology for bat carcass drag coefficient estimation. Field investigation at Macksburg wind farm resulted in the discovery of three bat species: Eastern Red bat (*Lasiurus borealis*), Hoary bat (*Lasiurus cinereus*) and Evening bat (*Nycticeius humeralis*). Carcass drop
experiments were performed from a dropping platform at finite height and carcass position time series data was recorded using a high-speed camera. Falling carcasses were subjected to aerodynamic drag and gravitational force. Carcasses were observed to undergo rotation; often rotating around multiple axes simultaneously and lateral translation. The carcass complex fall dynamics along with drop from limited height prohibits it from attaining the terminal velocity. Under this limitation, drag coefficient can be estimated by fitting ballistic model to the measured data. A new multivariable
optimization algorithm was performed to find the best-fit of the ballistic model to the measured data resulting in an optimized drag coefficient estimate. Sensitivity analysis demonstrated significant variation in drag coefficient with small a change in initial position highlighting the chaotic nature of carcass fall dynamics. Based on the limited sampling, the bat carcass drag coefficient range was found to be between 0.70 – 1.23.

## 1 Introduction and Review

Wind energy has become an increasingly important component of renewable energy (Pasqualetti et al. 2004; GAO 2005). As wind energy production has risen in many countries, it is giving rise to unexpected wildlife conservation issues (Morrison and Sinclair 2004). Wind turbines have been imposing risk to bats, as the result of collisions with turbine blades and associated infrastructure (Howell and DiDonato 1991). Hayes 2013 estimated that in 2012 alone, over 600,000 bats died as the result of interactions with wind turbines in wind energy facilities in the contiguous United States. In fact, bat mortality
has been reported at every wind energy facility studied to date (GAO 2005; Kingsley and Whittam 2007; Kunz et al. 2007;





Kuvlesky et al. 2007; NAS 2007; Arnett et al. 2016). It is not possible to prevent all bat deaths caused by wind turbines. However, bats in the United States Fish and Wildlife Service's (USFWS) Threatened and Endangered Species list must be monitored, and the wind energy facilities permitted, to ensure that allowable mortality rates are not exceeded.

The USFWS requires wind farm operators to perform carcass surveys within a specified radius around wind turbines to estimate the bat carcass take. However, guidance for the prescribed search radius around turbines is based on limited data. This could lead to surveys conducted where bats are unlikely to be found, or alternatively, limited search areas could miss bat carcasses outside the survey area. Turbine operators need a reliable method to guide their survey efforts and to determine the appropriate extent of the surveys and the most likely locations where bat carcasses can be found around the turbines. A technically defensible survey is critical to help operators determine whether wind turbines adversely affect listed
species and to evaluate project impacts.

Figure 1 shows the schematic of the three-blade horizontal axis turbine with its important components such as tower, hub, rotor blades, hub height and rotor swept area. The vertical distance from the ground to the turbine hub is called the hub height. In Fig. 1, three rotor blades are denoted in grey colour and the red dotted periphery indicates the rotor swept area.

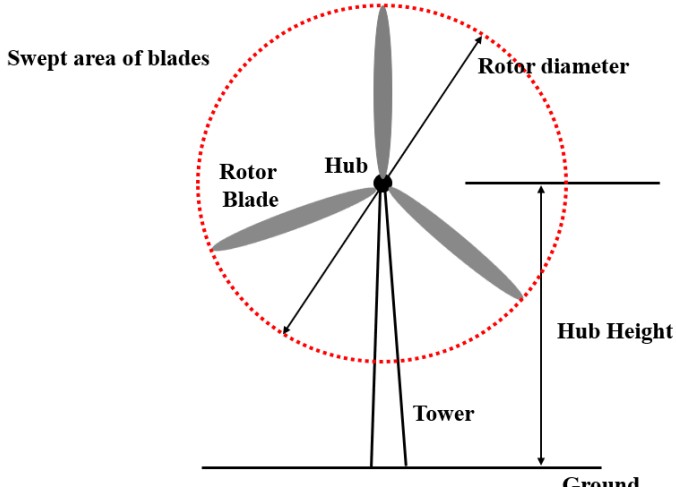

**Fig. 1: Three-blade horizontal axis wind turbine**


Few studies (Arnett 2005; Smallwood and Thelander 2005) have suggested the size of fall zone for birds and bats to estimate the carcass search area. Osborn et al. 2000 quantified the search area by dropping carcasses from the nacelle and the upper and lower bounds of the rotor-swept area on days with the brisk wind. But this method did not consider the effect of impact with moving blades on carcass fall trajectory. Gauthreaux 1996 suggested that the search area should be circular, with
a minimum radius proportional to the height of the turbine. He suggested the search area to be within 70 m of the turbine. Thelander and Runge 2000 found the average fall distance of birds to be 20.2 m, with 75% of birds falling less than 30 m



away from the tower. It is not clear whether some of these studies has any bias in search radius estimates due to insufficient search zones. Smallwood 2007 mentioned that inadequate search radius could cause bias in the carcass survey data.

Huso and Dalthorp 2014 proposed polynomial logistic regression models of relative carcass density as a function of distance from the nearest turbine. The study considered the carcass search locations at a number of turbine sites in Philadelphia: 15 turbines at Locust Ridge in 2010, and 22 turbines and 15 turbines at Casselman, in 2008 and 2011 respectively. The best-fit logistic model of carcass densities was found to be cubic for Casselman and linear for Locust Ridge. This study limited the search area for bat carcasses to 80 m. If the bat carcasses land beyond the 80 m distance, the surveyor misses those bat carcasses.

Hull and Muir 2010 (to be referred to as HM10) combined the Monte-Carlo approach with ballistic theory to propose a model that estimates the fall zone of different sized bird and bat carcasses after they are hit by different sized turbines. The mechanics-based ballistic model describes the trajectory of the bat carcass by relating the variation in fall velocity to the net resultant forces on a carcass, which include gravitational and aerodynamic drag forces. HM10 employed a fourth order Runge-Kutta (RK4) (e.g., Chapra and Canale 1988) method to numerically integrate the ballistic model to

determine the position and velocity of the carcass relative to the turbine base at each time step.

    HM10 assumed that the bats would be incapacitated after the strike and unable to affect their trajectory. HM10 assumed that the carcasses would be stationary in the rotor plane before being hit by the blade, which means the authors did not account for any initial pre-collision velocity. HM10 also assumed calm conditions with no turbulence, resulting in no wind drift effects on the carcass' fall. HM10 assumed the equal likelihood of strike anywhere in the rotor-swept area. HM10

modelled the carcass as a tumbling object, allowing the projected area and drag coefficient ($C_d$) to change randomly during the fall. HM10 performed simulations with the coefficient of restitution ($e$) being zero. This corresponds to the case of fully inelastic collision in which the restoration impulse is equal to zero. HM10 proposed a simple multiple linear regression model considering hub height and rotor radius as input variables to predict the carcass search radius.

    The maximum distance a bat carcass falls away from the base of a turbine after collision with a rotating blade is

governed by a number of factors, but arguably the most uncertain are the carcass aerodynamic characteristics. To compute the drag force, HM10 assumed $C_d$ for bat carcasses lie between 0.875 – 1.125, with a mean value of 1.00. The drag coefficient is a measure of the effectiveness of a streamlined object in reducing the fluid resistance faced by the object motion. Low drag coefficient means that the streamlined shaped object is enable to move easily through the ambient fluid due to minimum resistance whereas high drag coefficient implies the poor streamlining of the object causing the high

resistance to motion.

    Norberg 1976 suggested a $C_d$ range of 0.4 – 1.2 for a flying long-eared bat. But this is applicable to live bats for a single species only. No literature is available on $C_d$ for bat carcasses of different species. HM10 mentioned that little evidence exists to understand the aerodynamic characteristics of an injured animal in flight. Hedenström and Liechti 2001 estimated $C_d$ of passerine birds by measuring their dive speeds. They stated that some passerine birds terminate their

migration by diving abruptly toward the ground to land. Hedenström and Liechti 2001 measured this type of dive by tracking



migratory birds with radar. They maintained a tracking time long enough to allow the diving birds to attain the terminal velocity (no net acceleration). In this situation, drag coefficient can be calculated by balancing drag force with the gravitational pull. Based on measurements for 39 cases of diving birds, drag coefficient was estimated to be 0.37±0.13.

Given the lack of $C_d$ measurements of bat carcasses and the large range of previously reported $C_d$ values (0.4 – 1.2)
limited to a specific species in flight, the modelled carcass fall distribution histogram and maximum fall distance is highly uncertain. A reliable estimate of $C_d$ for individual species that considers carcass mass, size and shape is needed for modelling carcass fall trajectories and to determine the maximum fall distance. The main objective of the present study is to estimate the empirical drag coefficient ($C_d$) of bat carcasses by:

1. Performing bat carcass drop experiments to acquire time vs. position data,
2. Implementing the ballistic theory to estimate carcass drag coefficient.

The paper is organized as follows: Methodology section which includes research design for carcass drag coefficient estimation, ballistic model description, experimental set up, its various components, data acquisition procedure, limitations of the measured data and the newly suggested $C_d$ estimation algorithm. After methodology, there is results section discussing the drag coefficient estimates and the sensitivity analysis. The manuscript ends with the summary and conclusion section.

## 2 Methodology

### 2.1 Research Design for Drag Coefficient Estimation

Following methodology is proposed to compute the carcass drag coefficient of which no measurement is available:

1. Collect fresh bat carcass and perform the carcass drop experiment to acquire the time vs. position data during the fall.
2. Check if the carcass attain terminal velocity during the fall. If carcass attain terminal velocity; calculate drag coefficient by equating the drag force to the gravitational force.
3. If carcass don't attain the terminal velocity, find carcass drag coefficient by finding the best-fit of the ballistic model to the measured position or velocity data.

### 2.2 Ballistic Model Description

Projectile motion considers the influence of gravity only and neglects fluid resistance. The closed-form solution of projectile motion can be obtained easily. The Ballistic model describes the fundamental theory of projectile trajectories based on quadratic drag model accounting for the effect of fluid resistance. Ballistic model encompasses more realistic case and much harder to deal with. HM10 implemented following set of equations describing the ballistic theory for the velocity and acceleration of a carcass:




$$\frac{d\boldsymbol{x_p}}{dt} = \boldsymbol{u_p}$$

$$\frac{d\boldsymbol{u_p}}{dt} = -\frac{\rho_f C_d A_p |\boldsymbol{u_p} - \boldsymbol{u_f}|(\boldsymbol{u_p} - \boldsymbol{u_f})}{2m_p} - \boldsymbol{g} \tag{1}$$

where $m_p$ is carcass mass, $A_p$ is carcass projected area, $C_d$ is carcass drag coefficient, $\boldsymbol{u_f}$ is the fluid velocity vector, $\boldsymbol{u_p}$ is the carcass velocity vector, $\boldsymbol{x_p}$ is the fall position of carcass with respect to turbine base, $\rho_f$ is the fluid density and $\boldsymbol{g}$ is gravitational acceleration vector. The equation of motion described by Eq. (1) are coupled nonlinear equations. An exact solution of one-dimensional ballistic model for an isotropic object falling from rest in quiescent flow condition can be

obtained relatively easily. For any object falling along vertical direction ($z$) from rest in quiescent flow, Eq. (1) can be rewritten in the following form (upward (+) and downward (-)):

$$\frac{dw}{dt} = \frac{\rho_f C_d A_p w^2}{2m_p} - g \tag{2}$$

By integrating Eq. (1), the exact expression for carcass instantaneous velocity ($w(t)$) with zero initial velocity ($w$ at ($t = 0$) = 0) can be expressed as:

$$w(t) = w_t \tanh\left(\frac{-gt}{w_t}\right) \tag{3}$$

where $w_t = (2m_p g / \rho_f C_d A_p)^{1/2}$ is the terminal velocity attained by the carcass under dynamic equilibrium condition and $w(t)$ is the instantaneous velocity. By integrating Eq. (3) with respect to time, the exact solution for carcass instantaneous position, $z$ ($t$), can be obtained as:

$$z(t) = z_0 - \frac{w_t^2}{g} \ln\left(\cosh\left(\frac{-gt}{w_t}\right)\right) \tag{4}$$

where, $z_0$ is the carcass drop height at time $t = 0$.

Figure 2 represents the time vs. velocity plot obtained from the analytical solution of the ballistic model (Eq. (3)) by

considering the mean values of $m_p = 14$ g, $A_p = 28$ cm$^2$ and $C_d = 1$ for bat carcass from HM10. The acceleration curve in Fig. 2 can be explained via following three phases of fall:

(a) Initial phase: In this phase, carcass just started gaining small magnitude of velocity under gravity. As a result, the resistive drag force is quite small as compared to the gravitational force. This stage of fall is represented by the straight-line region in Fig. 2 highlighting the net acceleration being nearly constant.





(b) Terminal velocity phase: In this phase, carcass had attained the terminal velocity, i.e., drag force equals the gravitational force. This feature in Fig. 2 is evident by the flat portion of the curve when the carcass velocity becomes constant during the later stages of the acceleration curve.

    (c) Transition phase: In this phase, the resistive drag force gains more strength because of increase in velocity and eventually equals the gravitational force to attain the terminal velocity (Highlighted by red circle in Fig. 2). The

transition phase describes the fall dynamics between the gravity dominant phase and terminal velocity phase in an exponential manner. The magnitude of the resistive drag force will be governed by the empirical drag coefficient ($C_d$) of bat carcasses for which there is no measurement.

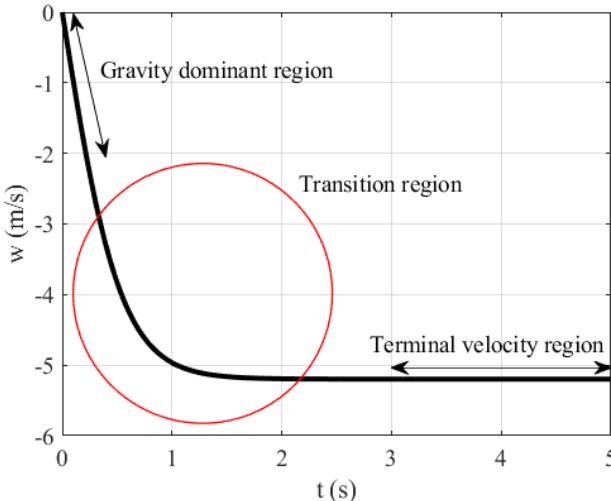

**Fig. 2: Acceleration curve from exact solution of the ballistic model**

**2.3 Experimental Equipment, Materials and Procedures**

The ballistic model is useful for guiding carcass surveys only if it can accurately predict carcass fall trajectories. This can only be done if the aerodynamics of the carcasses are known. Given the lack of measurements available for bat carcass $C_d$, bat carcass drop experiments at a wind farm were performed and recorded by a high-speed camera. Biologists discovered following three species of bat carcasses in Macksburg while conducting post-construction surveys on the day of the experiment: Eastern Red bat (*Lasiurus borealis*), Hoary bat (*Lasiurus cinereus*), and Evening bat (*Nycticeius humeralis*).

Macksburg wind farm is operated by the MidAmerican Energy Cooperation (MEC) in Iowa. It has 52 Siemens SWT – 2.3 MW wind turbines. The radius of turbine rotor is 54 m and turbine hub height is 80 m. The bat carcasses' mass, length (excluding tail, *a*), and lateral dimension (*b*) was measured using a weighing scale and ruler, respectively (Fig. 3). For bat carcasses, the lateral dimensions orthogonal to length were assumed to be equal which means *b* = *c*. The irregularly shaped





bat carcasses were approximated as an ellipsoid and based on it, its volume equivalent diameter ($d_{eq}$) (Mandø and Rosendahl

2010) was calculated in order to have a geometric representation of the bat carcass in the ballistic model. The variable $d_{eq}$

represents the diameter of the sphere with the same volume as the irregularly shaped bat carcass.

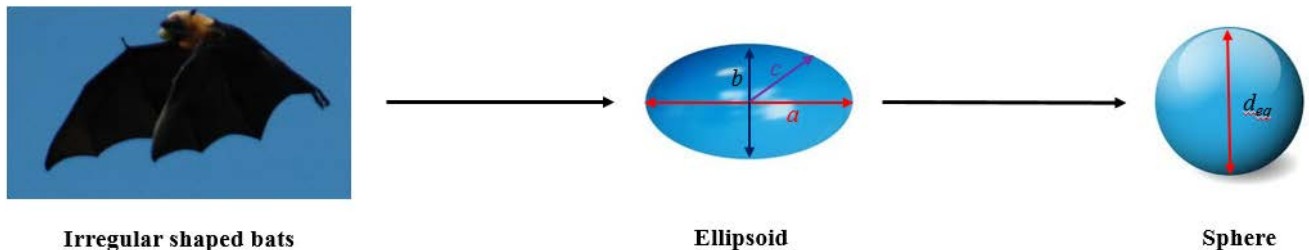

Irregular shaped bats                          Ellipsoid                          Sphere

**Fig. 3: Representation of bats in the ballistic model. The bat image in the figure is procured from Creative Commons**

**(https://ccsearch.creativecommons.org/photos/cefc843b-50c2-4651-af3f-e1bf5af75311).**

Table 1 lists the mass, body dimensions, and equivalent diameter of bat carcasses searched on the day of experiments.

**Table 1: Physical properties of bats**

| Species | Mass (g) | Length ($a$, cm) | Lateral Dimensions ($b$ & $c$, cm) | $d_{eq}$ (cm) |
|---|---|---|---|---|
| Hoary Bat | 24 | 7.6 | 3.8 | 4.8 |
| Eastern Red Bat | 9.7 | 5.0 | 2.5 | 3.2 |
| Evening Bat | 1.5 | 3.8 | 1.9 | 2.4 |

Figure 4 shows an annotated photo of the bat carcass drop experiment. Freshly collected carcasses were dropped from a

finite initial height ($z_0$) in front of a 6.3 m high wall. For each species, two experiments were performed and recorded using a

high-speed camera to extract the carcass position. The wall was marked using horizontal strips of tape over a total distance

of 4.5 m. The side view of the entire experimental set-up geometry is illustrated in Fig. 5, along with experiment components

such as the wall with markings, the camera mounted on a tripod and its field of view, and the carcass dropping platform. The

carcass drop was performed from a dropping platform set at 1 m in front of the wall ($y_2$ in Fig. 5). High-speed camera of

Integrated Design Tools Inc. (IDT, NX4-S2 model) located in Pasadena, California was used to record the carcass drop

experiments. The camera mounted on a tripod was positioned at 18 m distance from the wall (($y_1+y_2$), as seen in Fig. 5). The

array size of the images acquired by the high-speed camera was 1024 × 1024 pixel. The markings on the wall were used to

calibrate the images and size of a single pixel in the images was determined to be 7.1 mm. The camera records at 500 frames

per second and therefore has a temporal resolution of 0.002 s.




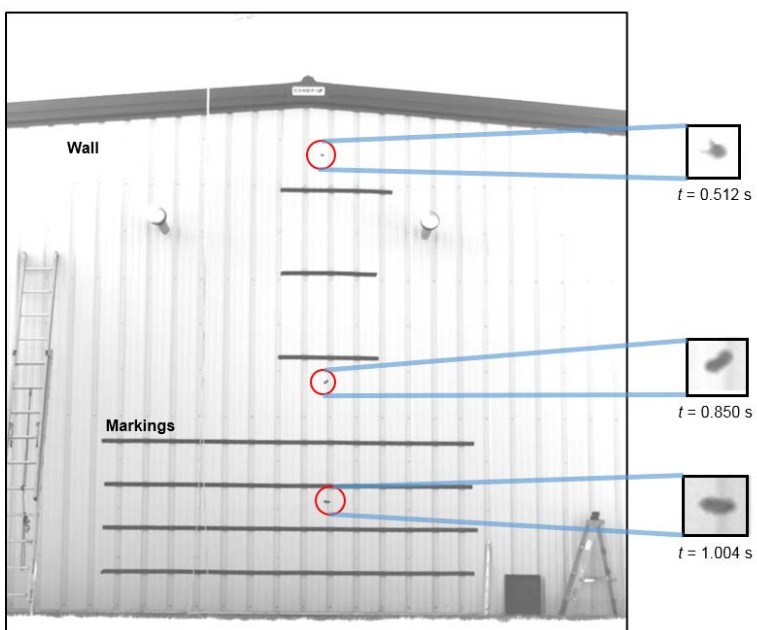

**Fig. 4: Hoary bat carcass drop experiment**

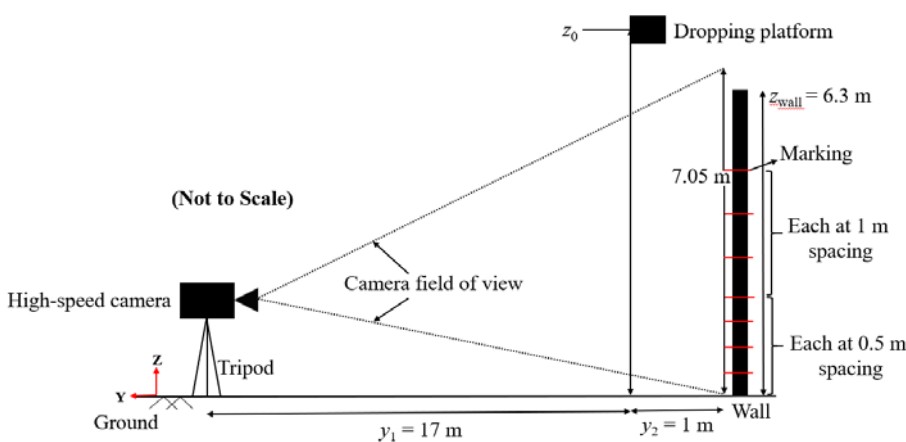

**Fig. 5: Side view of carcass drop experimental set-up**

The images obtained from the high-speed video recording were used to determine the vertical position ($z$) of the bat

carcass at a specific instant. Bat carcasses are of irregular shape and have finite size. Motion Studio X64 software which



includes applications for the operation of all IDT high-speed digital cameras, was used to extract the carcass pixel information from a particular image frame. At $t = 0.722$ s, the Hoary bat carcass top and bottom edge pixel value in $z$-direction, i.e., $(z_{pixel})_{top}$ and $(z_{pixel})_{bottom}$ were found to be 448 and 454 respectively (Fig. 6 (a)). The arithmetic mean of $(z_{pixel})_{top}$ and $(z_{pixel})_{bottom}$ values is 451, which when multiplied by 7.1 mm gives the carcass position with respect to the ground

($z$) as 3.86 m. This procedure was further carried out for every alternate image frame obtained during carcass drop experiment to generate a time vs. position time series for the Hoary bat drop with time interval $\Delta t = 0.004$ s (Fig. 6 (b)).

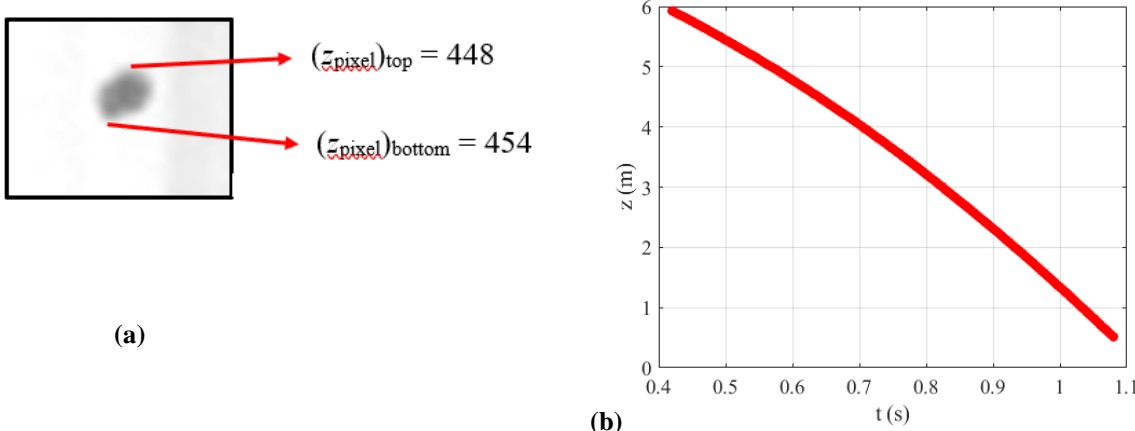

**Fig. 6: (a) Position extraction for Hoary bat; (b) Time vs. measured position**

### 2.4 Velocity ($w$) estimates from Measured Position ($z$)

The time vs. position information extracted from high-speed imaging was used to calculate the fall velocity ($w$) using the central-differencing numerical scheme (Chapra and Canale 1988). The scheme formulation for vertical velocity at $i^{th}$ time

instant is given as:

$$w(i) = \frac{z(i + 1) - z(i - 1)}{2(\Delta t)} \tag{5}$$

where $z$ ($i$-1) and $z$ ($i$+1) represent the carcass position at $(i–1)^{th}$ and $(i+1)^{th}$ time, respectively. Figure 7 shows the velocity computations obtained by applying the central-differencing scheme on the measured position data. The falling objects reach their terminal velocity, $w_t$, when the force of gravity is balanced by the aerodynamic drag force. However, because of the limited drop height, none of the carcasses attained terminal velocity during the experiments, as the data points of the

measured velocity lie nowhere close to the terminal velocity region shown in Fig. 7. Since, the bat carcasses did not achieve terminal velocity during the experiments, it is not possible to calculate $C_d$ by equating the drag force with the gravitational force. However, with the assumption of terminal velocity attainment at the end of the carcass fall trajectory, the drag and





gravitational force can be equated to each other, in order to compute carcass $C_d$. The instantaneous velocity estimate at the end of a particular carcass drop experiment and corresponding $C_d$ for the three discovered species are mentioned in Table 2.

**Table 2: Drag coefficient computed from the terminal velocity assumption**

| Species | Velocity (m/s) | Drag coefficient |
|---|---|---|
| Hoary bat | 9.76 | 2.27 |
| Eastern Red bat | 7.97 | 3.10 |
| Evening bat | 7.1 | 1.10 |

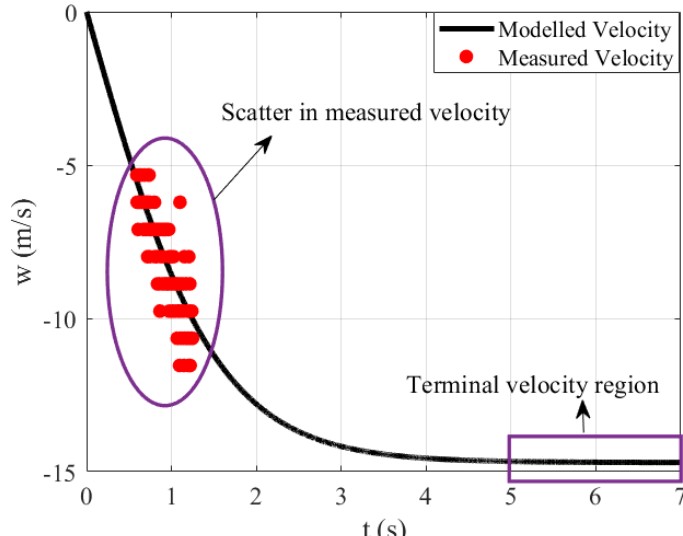

**Fig. 7: Time vs. measured velocity**

In the absence of the terminal velocity attainment, $C_d$ needs to be estimated by finding the ballistic model's best-fit to the measured position or velocity. It will be interesting to see the difference between the $C_d$ values obtained from the terminal velocity assumption at the end of the fall (Table 2) and from the best-fit of the ballistic model to the measured data. However, the velocity computed from the measured position at temporal resolution of 0.004 s has significant scatter (Fig. 7). On careful observation of measured velocity, one can notice the stratification in the velocity values which means that measured velocity attains few selective values only. This phenomenon is the result of the peak-locking error, which is the most significant bias error (Chen and Katz 2004). Peak-locking is the systematic tendency of the measured particle image displacement to be biased toward the closest integer pixel value (Westerweel 1997). It arises when the object size is either smaller or larger than the smallest pixel size in the image. In this situation, the object displacement for a very small-time interval is locked within a single pixel. It is an experimental method problem that can be resolved either by allowing significant object displacement across the pixels or by obtaining sub-pixel displacement. In the present case, the peak-



locking error is eliminated by coarsening (filtering) the raw position data at an appropriate coarsening window ($\Delta t_c$) to obtain a scatter-free measured velocity dataset.

While extracting the position from the high-speed images, it was noticed that the carcasses were rotating — sometimes with simultaneous rotation around multiple axes. Evening bat was observed to experience lateral translation, in addition to the above-mentioned complex features of fall dynamics. Fig. 4 highlights snapshots of the Hoary bat carcass' numerous orientation features at different time instants. These characteristics emerge because of carcass shape asymmetry and cause a change in the carcass' aerodynamics during the drop experiments from limited height, which ultimately prevents it from attaining the terminal velocity. The carcass fall dynamics' complex traits can be averaged by selecting the measured

position values over an appropriate filtering window ($\Delta t_c$). The scatter in measured velocity is caused primarily due to peak-locking error in acquired images. An improved analysis methodology is required to obtain robust estimates of the carcass drag coefficient.

## 2.5 New methodology for carcass drag coefficient estimation

The ballistic model defined by Eq. (1) is an initial value problem, which means the initial condition for position ($z_0$) and the

velocity ($w_0$) is required to solve it analytically. Unfortunately, the hand of the person dropping the carcass was not visible in the recorded images. Therefore, $z_0$ and $w_0$ could not be determined. The lack of accurate information about $z_0$ and $w_0$ generated two more unknown variables: $z_0$ and $w_0$. As discussed earlier in section 2.4, the measured velocity data obtained through high-speed imaging has significant scatter, which needs to be filtered to obtain scatter-free dataset. This scatter-free data can be fitted to the ballistic model to compute robust $C_d$. Thus, the process of finding the bat carcasses' drag coefficient,

$C_d$, turned out to be a multivariable optimization problem. This can be framed in the following question: What is the optimal resolution ($\Delta t_c$) of the measured data giving the best-fit to the ballistic model with optimized estimates of $z_0$, $w_0$, and $C_d$? Following algorithm was introduced to estimate bat carcass $C_d$:

1. With the assumption $w_0 = 0$, one degree of freedom in the problem is reduced, leaving three degrees of freedom ($\Delta t_c$, $z_0$, $C_d$).

2. An array of plausible $z_0$ and $C_d$ values is declared, and then the ballistic model was solved analytically for the prescribed values of $z_0$ and $C_d$. The $z_0$ array with $m$ elements and $C_d$ array with $n$ elements results in $m \times n$ fall trajectories.

3. An array representing different resolutions, $\Delta t$, of position data with $p$ elements is defined and the measured position data at these different resolutions is fitted to the $m \times n$ fall trajectories. This leads to $m \times n$ number of

$z_0$-$C_d$ combinations of fall trajectories fitted to $p$ different resolutions of measured data.

4. For each of the $m \times n \times p$ fitting events, Root Mean Square Error in fall velocity ($RMSE_w$) at i[th] time instant is estimated using the following formula:



$$RMSE_w = \sqrt{\frac{\sum_{i=1}^{n}\left(w_{model}(i) - w_{field}(i)\right)^2}{n}} \tag{6}$$

where $w_{model}$ represents the velocity obtained from the exact solution of the ballistic model and $w_{field}$ represents
the measured velocity from high-speed imaging of carcass drop experiments. The variable $n$ is the number of
data points in velocity time series for a specific coarsening window.

5.   For $m \times n$ combinations of $z_0$ and $C_d$, $\Delta t$ vs. $RMSE_w$ is plotted. The range of $\Delta t$ values over which $RMSE_w$
remains invariant for $z_0$-$C_d$ combinations is identified. The basis of selecting $\Delta t$ range of invariant $RMSE_w$ is:
the relative error in successive $RMSE_w$ values being less than 10%.

$$RELError(i) = \left(\frac{RMSE_w(i+1) - RMSE_w(i)}{RMSE_w(i)}\right) \times 100 \tag{7}$$

where $RMSE_w$ (i+1) and $RMSE_w$ (i) are the two successive values in $RMSE_w$ vector at (i+1)[th] and i[th] time step
for a specific $z_0$-$C_d$ combination. $RELError$ (i) is the relative error in $RMSE_w$ at i[th] time step.

6.   For each element in $\Delta t$ vector (which corresponds to invariant $RMSE_w$) representing a plausible optimum
resolution of the measured data, the optimal $z_0$ and $C_d$ is computed by defining $(RMSE_w)_{min}$ as the objective
function. The temporal resolution of extracted data yields serious scatter which makes it impossible to find the
best-fit of ballistic model to this data. For large filtering window of the measured data, the order of the ballistic
model becomes equal to the number of the data points; hence giving the biased estimate of $RMSE_w$. The
objective for selecting the $\Delta t$ range of constant $RMSE_w$ is to have an unbiased estimator of the goodness of fit
and therefore, range of $\Delta t$ yielding invariant $RMSE_w$ is selected as possible candidate for the optimal coarsening
window.

7.   From step 6, a pool of initial positions ($z_0$) and drag coefficients ($C_d$) for varying resolutions of measured data
(embedded in $\Delta t$ vector corresponding to constant $(RMSE_w)_{min}$) is obtained. Out of this pool, the value of $\Delta t$, $z_0$
and $C_d$ giving the global minimum $(RMSE_w)_{min}$ is selected as optimum data resolution, initial position, and
carcass drag coefficient.

8.   To test the accuracy of the optimum $\Delta t_c$, $z_0$ and $C_d$, the analytical solution of the ballistic model is compared
with the measured position and velocity at the optimum resolution ($\Delta t_c$). The measured velocity is obtained by
applying central difference finite difference on the measured position.



## 3 Results

The newly proposed $C_d$ estimation algorithm based on multivariable optimization was applied to the measured position data of the bat carcass drop experiments. Following are the results for the three bat species:

### 3.1 Hoary bat

For carcass drop experiment, $z_0$ was defined between 7.40 m and 7.80 m at an increment of 0.01 m whereas $C_d$ array was selected between 0.50 and 1 with differential $C_d$ being 0.01. In this manner, there are overall $41 \times 51$ $z_0$-$C_d$ combinations leading to equal number of carcass fall trajectories. The $\Delta t$ array was chosen from 0.004 s to 0.512 s at an increment of 0.004 s. This declaration of $z_0$-$C_d$-$\Delta t$ culminated in $41 \times 51 \times 128$ ballistic model fitting events to the measured data. For each of these cases, $RMSE_w$ was calculated through difference in the modelled velocity and measured velocity values.

Three-point centered moving average of $RMSE_w$ vector was computed and plotted with $\Delta t$ (for all $z_0$-$C_d$ combinations) in order to identify $\Delta t$ range of invariant $RMSE_w$ on the basis of $RELError < 10\%$. Figure 8 (a) demonstrates the $\Delta t$ vs. $RMSE_w$ plot, for the lower ($z_0 = 7.40$ m and $C_d = 0.50$) and upper ($z_0 = 7.80$ m and $C_d = 1.00$) bounds of $z_0$ and $C_d$ array respectively. The $\Delta t$ range corresponding to invariant $RMSE_w$, was found to be between 0.060 and 0.104 s (region between the two vertical arrows in Fig. 8 (a)). Optimum $z_0$ and $C_d$ were calculated by minimizing $RMSE_w$ for each element in $\Delta t$ vector corresponding to invariant $RMSE_w$. Ultimately, the global minimum $(RMSE_w)_{min}$ was selected as a criterion for $\Delta t_c$, $z_0$ and $C_d$ in highlighted spectrum of $\Delta t$ giving invariant $RMSE_w$.

Fig. 8 (b) shows the plot of $RMSE_w$ in the $z_0$-$C_d$ plane at an optimal resolution of $\Delta t_c = 0.104$ s, in highlighted domain of $\Delta t$ vector. The red dot in the heatmap displays the optimum $z_0 = 7.58$ m and $C_d = 0.70$ yielding global minimum $(RMSE_w)_{min}$ of 0.0666 m/s. The accuracy of optimal $z_0$ and $C_d$ was tested by comparing the measured position and velocity at the optimal filtering window, $\Delta t_c$, to the analytical solution of position and velocity. Figure 9 displays the good affinity between the measured position and velocity with exact expression of position and velocity, using the computed $z_0$ and $C_d$ values.



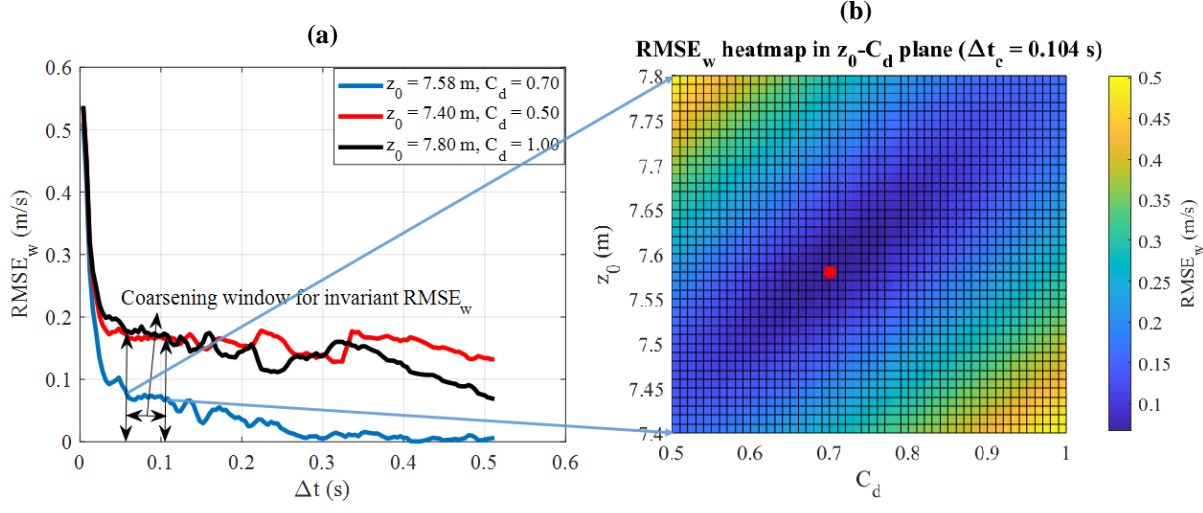

**Fig. 8: (a) Δt vs. *RMSE*�ww plot; (b) *RMSE*ww heatmap in $z_0$-$C_d$ plane (Hoary bat)**

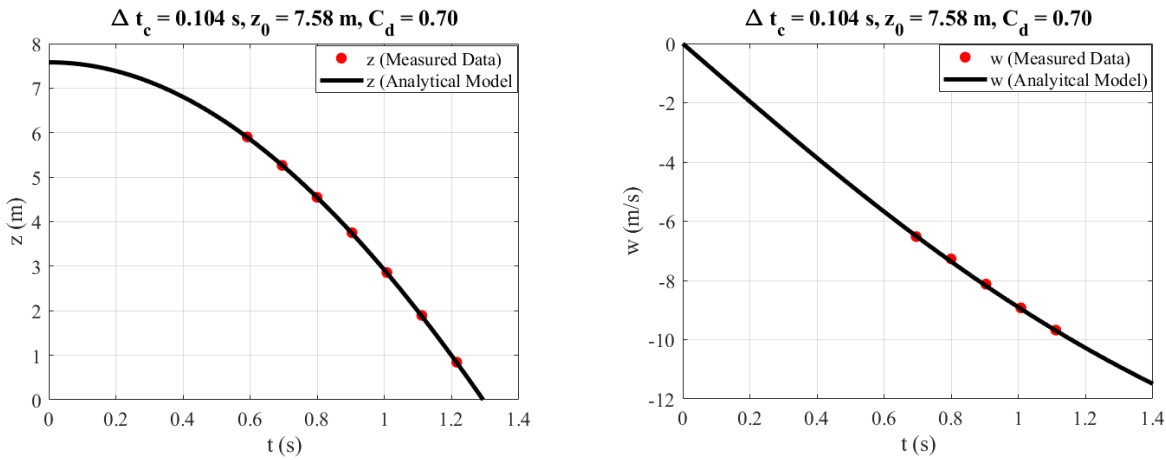

**Fig. 9: Comparison of position and velocity for Hoary bat (analytical solution vs. measured data)**

### 3.2 Eastern Red bat

For analysing Eastern Red bat, $z_0$, $C_d$ and $\Delta t$ were defined in the same manner as with the Hoary bat. This declaration of $z_0$-$C_d$-$\Delta t$ culminated in $41 \times 51 \times 128$ ballistic model fitting events to the measured data. For each of the cases, $RMSE_w$ was calculated through difference in the modelled velocity and measured velocity values. Again, three-point centered moving average of $RMSE_w$ vector was computed and plotted with $\Delta t$ (for all $z_0$-$C_d$ combinations) in order to identify $\Delta t$ range of invariant $RMSE_w$ on the basis of $RELError < 10\%$.



Figure 10 (a) demonstrates the $\Delta t$ vs. $RMSE_w$ plot, for the lower and upper bounds of $z_0$ and $C_d$ array respectively. The spectrum of $\Delta t$ corresponding to invariant $RMSE_w$ was found to be between 0.080 s and 0.152 s (region between the two vertical arrows in Fig. 10 (a)). For each element in the above-mentioned $\Delta t$ range of constant $RMSE_w$, the optimum $z_0$ and $C_d$ were calculated by minimizing $RMSE_w$ and then from this pool of $(RMSE_w)_{min}$, global minimum $(RMSE_w)_{min}$ was selected as

the criteria to identify $\Delta t_c$, optimized $z_0$ and $C_d$ for that specific carcass drop experiment.

Figure 10 (b) shows the plot of $RMSE_w$ in $z_0$-$C_d$ plane, at an optimum filtering of $\Delta t_c = 0.152$ s, in the marked range of $\Delta t$ vector. The optimized $z_0$ (7.63 m) and $C_d$ (0.80) corresponding to the global minimum $(RMSE_w)_{min}$ of 0.0445 m/s, is highlighted by red dot in Fig. 10 (b). For testing the accuracy of optimal $z_0$ and $C_d$ estimates, the measured position and velocity data at $\Delta t_c$ resolution was compared with the exact expression of position and velocity. It is evident from Fig. 11 that

the measured position and velocity data is in good agreement with the analytical expression of position and velocity with the optimized estimates of $\Delta t_c$, $z_0$ and $C_d$.

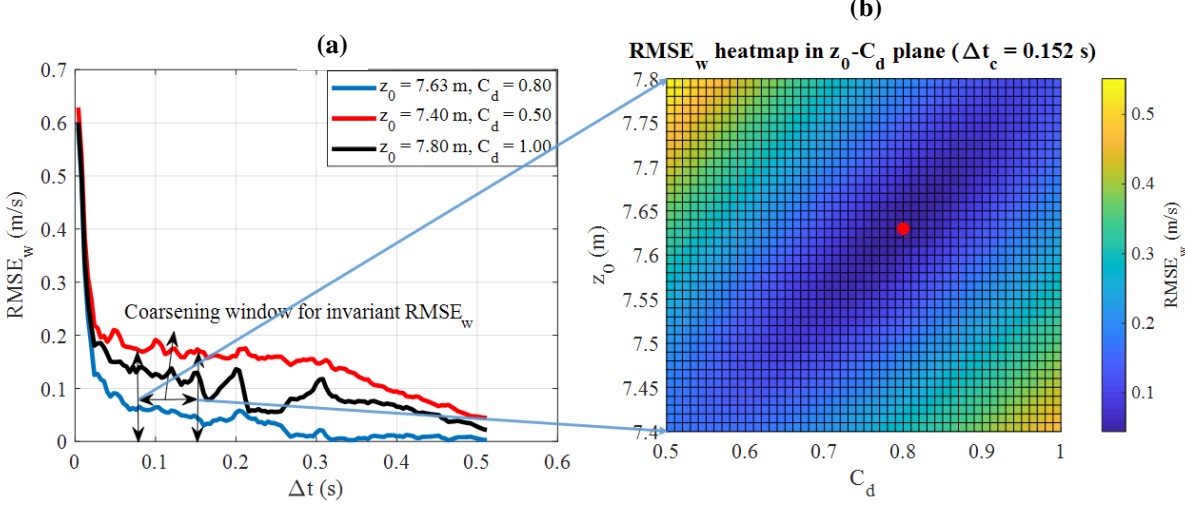

Fig. 10: (a) $\Delta t$ vs. $RMSE_w$ plot; (b) $RMSE_w$ heatmap in $z_0$-$C_d$ plane (Eastern Red bat)





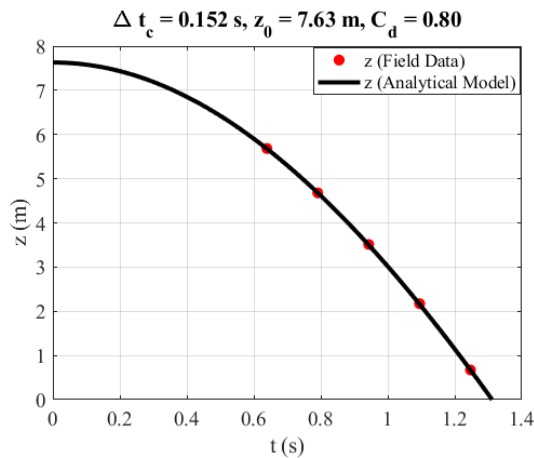 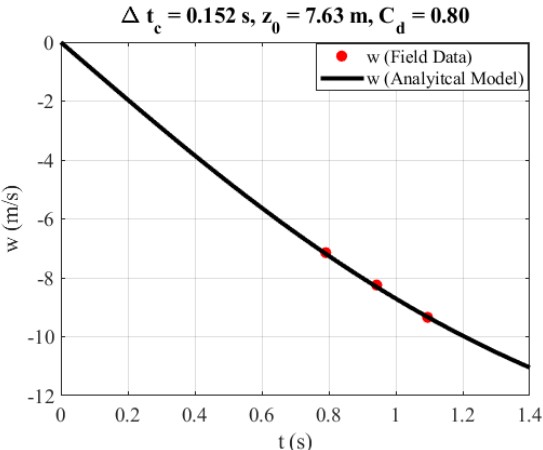

**Fig. 11: Comparison of position and velocity for Eastern Red bat (analytical solution vs. measured data)**

### 3.3 Evening bat

For Evening bat, $z_0$ array was declared between 6.90 m and 7.70 m with differential $z_0$ being 0.01 m whereas $C_d$ vector was defined between 0.90 and 1.20 at an increment of 0.01. $\Delta t$ array was kept the same as it was with Eastern Red bat and Hoary bat. This declaration generated overall $81 \times 31 \times 128$ ballistic model fitting events to the different filtering windows of measured data.

     Figure 12 (a) shows the $\Delta t$ vs. $RMSE_w$ (moving averaged) plot, for the lower and upper bounds of $z_0$ and $C_d$ array

respectively. $\Delta t$ range corresponding to invariant $RMSE_w$ was established between 0.132s and 0.144 s (region between vertical arrows in Fig. 12 (a)) and global minimum $(RMSE_w)_{min}$ was selected as a criterion for $\Delta t_c$, optimized $z_0$ and $C_d$, within marked range of $\Delta t$. The red dot in Fig. 12 (b) represents optimal values of $z_0 = 7.20$ m and $C_d = 1.01$ with global minimum $(RMSE_w)_{min}$ of 0.0777 m/s for $\Delta t_c = 0.144$ s. Figure 13 presents the comparison of the measured position and velocity with the analytical solution of position and velocity. The measured data was found to be in good agreement with the

expression indicating the accuracy of $z_0$ and $C_d$ estimates.



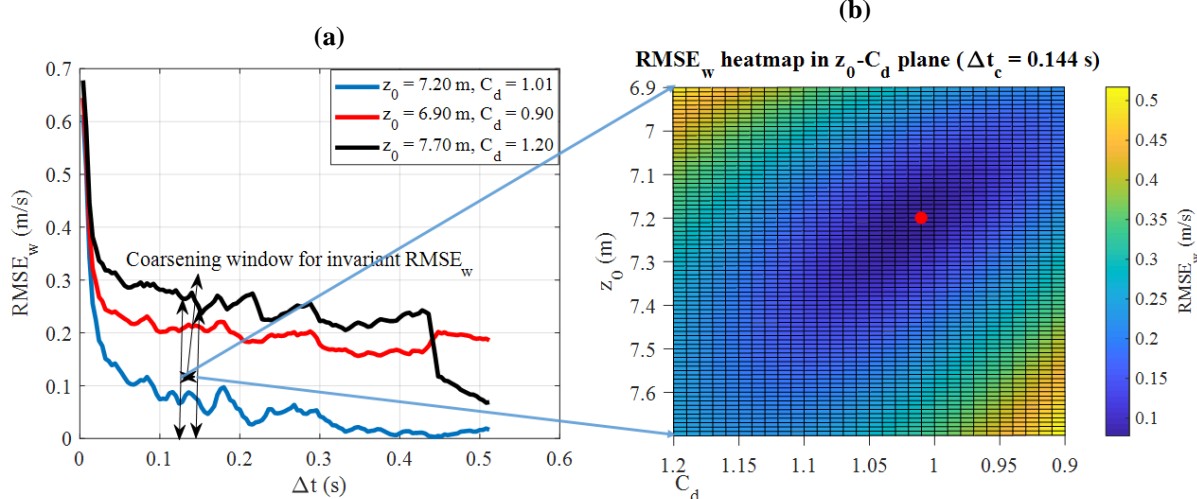

**Fig. 12: (a) Δt vs. *RMSE*ₔ plot; (b) *RMSE*ₔ heatmap in z₀-Cₔ plane (Evening bat)**

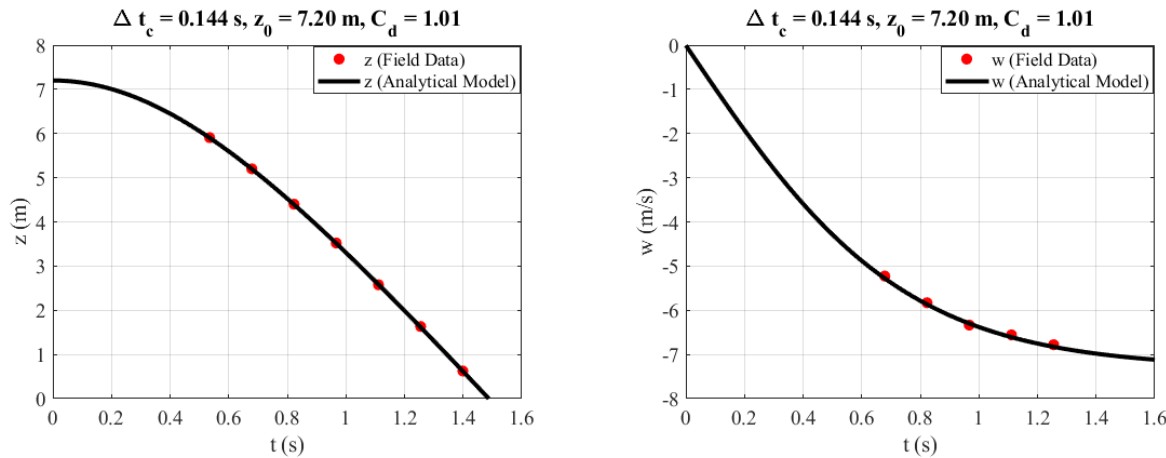

**Fig. 13: Comparison of position and velocity for Evening bat (analytical solution vs. measured data)**


Table 3 summarizes the optimal filtering window ($\Delta t_c$) of measured data, initial position ($z_0$) and drag coefficient ($C_d$), obtained via applying the multivariable optimization algorithm on the measured high-speed imaging data of carcass drop experiment for the three discovered species. On comparing $C_d$ values from Table 2 and Table 3, the significant difference in $C_d$ estimates of Hoary bat and Eastern Red bat from the two approaches are quite evident. It highlights the

incorrect assumption of terminal velocity attainment at end of the carcass drop experiment leading to overestimated $C_d$ values for these species. However, for Evening bat, the $C_d$ estimates from the two approaches were found to be close to each other indicating the possibility of Evening bat just entering into the terminal velocity phase during the last stages of fall.





**Table 3: Optimal filtering window, initial position and drag coefficient**

| Species | $\Delta t_c$ (s) | $z_0$ (m) | $C_d$ |
|---|---|---|---|
| Hoary bat | 0.104 s | 7.58 | 0.70 |
| Eastern Red bat | 0.152 s | 7.63 | 0.80 |
| Evening bat | 0.144 s | 7.20 | 1.01 |

## 4 Sensitivity of drag coefficient as a function of initial position

The sensitivity of $C_d$ with respect to $z_0$ is checked by perturbing the optimized $z_0$ by small amount (±1%) and observe the
percentage variation in optimized $C_d$ estimates (with earlier computed $\Delta t_c$). Table 4 presents the fluctuated $z_0$ values (column
2), corresponding $C_d$ values (column 3) and percentage difference in $C_d$ (column 4, considering optimized $C_d$ in section 3 as
the reference), for each bat species.

**Table 4: Drag coefficient sensitivity with respect to initial position**

| Species | Initial position ($z_0$) | Drag coefficient ($C_d$) | % difference in $C_d$ |
|---|---|---|---|
| Hoary bat | 7.66 m (+1%) | 0.79 | 13 |
| | 7.50 m (-1%) | 0.60 | 14 |
| Eastern Red bat | 7.71 m (+1%) | 0.85 | 6 |
| | 7.55 m (-1%) | 0.70 | 12 |
| Evening bat | 7.27 m (+1%) | 1.01 | 0 |
| | 7.13 m (-1%) | 0.99 | 2 |

From Table 4, it is noticeable that for Hoary and Eastern Red bat, even a small change of 1% in initial position is
capable of causing 6 – 14 % difference in $C_d$. It indicates the chaotic nature of the Hoary and Eastern Reed bat (heavy and
large) carcass fall dynamics when dropped from a limited height. However, in case of Evening bat (light and small), the
percentage change in $C_d$ is only 2% for 1% change in $z_0$. It is an important finding as 1% variation in $z_0$ corresponds to 7 cm
(approximately) which is of the order of $d_{eq}$ of larger species such as Hoary bat. So, depending upon the initial orientation of
carcass at $t = 0$, it is possible to have 1% difference in $z_0$ which further may lead to significant difference in carcass $C_d$.

## 5 Sensitivity of fall distribution histogram as a function of mass and drag coefficient

The $C_d$ range of the three discovered species from the two drop experiments was computed by implementing the newly
proposed $C_d$ estimation algorithm. Table 5 encapsulates the calculated $C_d$ range of Eastern Red bat, Hoary bat and Evening
bat from the two runs. On the basis of $C_d$ range of carcass of sampled species, the bat carcass drag coefficient was found to
be between 0.70 – 1.23. For Evening bat second drop experiment, the criteria of invariant $RMSE_w$ was selected as $RELError$
< 5%. This $C_d$ range of present study is different from the $C_d$ range of 0.875 – 1.125 used by HM10. In case of large





sampling of carcass of other bat species, there is a plausibility of bat carcass $C_d$ range being larger than the computed in the present study.

**Table 5: Drag coefficient range of bat carcasses**

| Species | Drag coefficient ($C_d$) |
| --- | --- |
| Hoary bat | 0.70 – 0.73 |
| Eastern Red bat | 0.74 – 0.80 |
| Evening bat | 1.01 – 1.23 |

The sensitivity analysis of carcass fall distribution histogram for Hoary bat and Evening bat was performed as a function of carcass mass and its drag coefficient. Hoary and Evening bat were selected for this exercise because they are the heaviest and lightest bat respectively. Figure 14 shows fall zone histogram variation for Hoary bat (upper row) and Evening bat (lower row) for the highest and lowest values of $C_d$ respectively. The bat population contains 3960 carcasses generated by uniform distribution of bats on the rotor plane, at radial resolution of 1 m and angular resolution of 5°. The rotor radius is 54 m and hub height is 80 m whereas the turbine RPM is 8.7 in the histogram calculations.

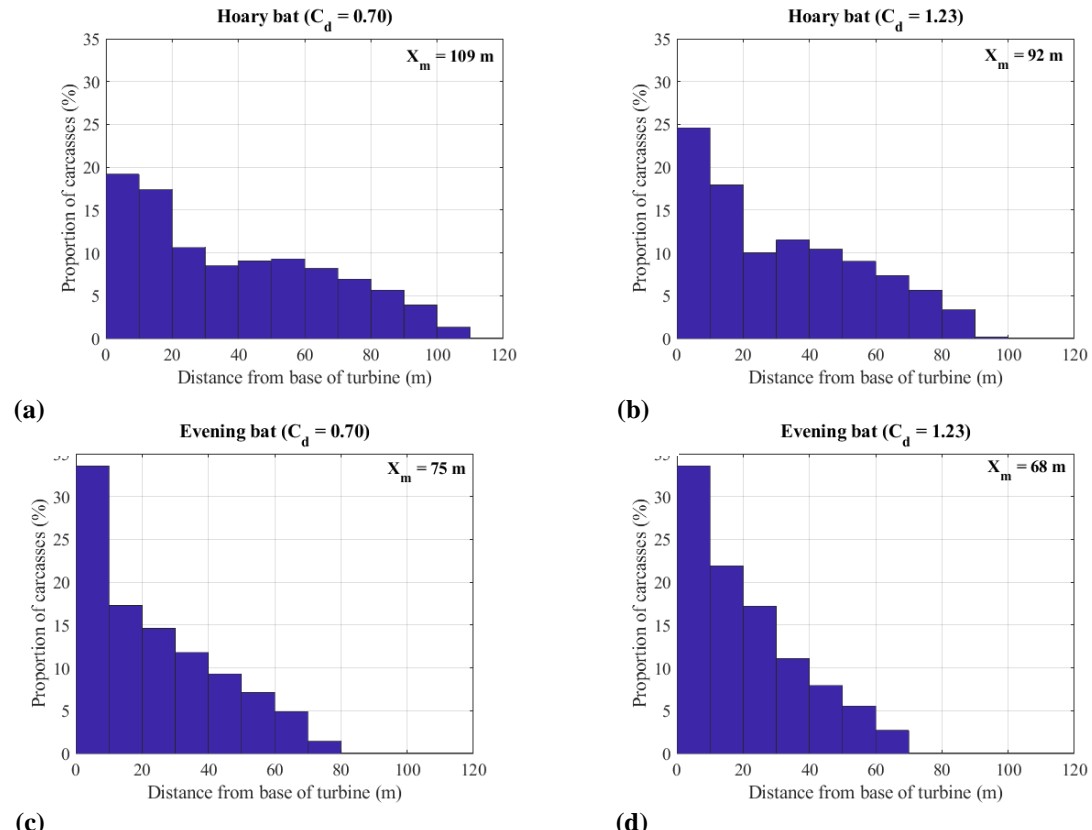

**Fig. 14: Carcass fall distribution histogram (a) Hoary bat ($C_d$ = 0.70) (b) Hoary bat ($C_d$ = 1.23) (c) Evening bat ($C_d$ = 0.70) (d) Evening bat ($C_d$ = 1.23)**






It is noticeable from Fig. 14 that for the same mass (or species), the maximum fall distance ($X_{max}$) varies significantly with $C_d$ variation. Increase in $C_d$ results in reduced value of $X_{max}$ for the same mass whereas decrease in mass leads to reduction in $X_{max}$ for the same $C_d$. It was found that heaviest bat with lowest $C_d$ covers maximum fall distance (109 m) and lightest bat with highest $C_d$ covers the minimum fall distance (68 m). The sensitivity analysis shown in Fig. 14

concludes the significance of drag coefficient in computing the upper bound on the maximum fall distance travelled by the bat carcasses using the ballistic theory.

## 6 Summary and Conclusions

The goal of this research was to make the first measurements of drag coefficient for bat carcasses. This data will allow for robust model of carcass fall distributions around wind turbines to guide carcass surveys. Fresh bat carcasses (Hoary bat,

Eastern Red bat and Evening bat) were discovered in the Macksburg wind farm to perform carcass drop experiments. Carcass fall trajectories were measured with high-speed video recording. Because of the carcass complex fall dynamics during the fall and limited drop height, the irregular shaped carcasses did not reach the terminal velocity. Therefore, $C_d$ needs to be estimated by finding the best-fit of the ballistic model to the measured position or velocity data. Being an initial value problem, ballistic model requires the initial position ($z_0$) and velocity ($w_0$) which could not be accurately recorded during the

high-speed imaging of the carcass drop experiments. The measured velocity data had significant scatter due to high temporal resolution resulting in no carcass movement in small time interval. This highlights the need to compute the appropriate filtering window ($\Delta t_c$) to deduce the scatter-free measured data to which the ballistic model can be fitted. For computing the above-mentioned unknowns ($\Delta t_c$, $z_0$, $w_0$, $C_d$), a multivariable optimization algorithm is proposed and implemented in a sequential manner yielding $C_d$ measurements of discovered bat carcasses in the field campaign. The limited sampling of bat

carcasses in wind farm resulted in the drag coefficient range of 0.70 – 1.23. The sensitivity of $C_d$ with respect to $z_0$ is tested by computing the percentage variation in $C_d$ with small change in $z_0$. It was found that for Hoary bat and Eastern Red bat, even a small difference of 1% (~7 mm) in $z_0$ yielded 6% - 14% (0.05 – 0.10) difference in $C_d$, indicating the high sensitivity of $C_d$ for heavy (large) bat carcass dropped from limited height.

**Code and data availability** Data and code can be made available upon request from the corresponding author.

**Author Contributions** SP conducted the study as part of his doctoral research supervised by CDM. CDM conceptualized the research, acquired financial support, and access to the equipment and facilities. SP conducted data analysis and produced final results, figures and original draft of the manuscript. SP and CDM collected the data, interpreted the analysis results, and

reviewed and revised the final version of the manuscript.



**Competing Interests** The authors declare no conflict of interest.

**Acknowledgements**

The study was conducted with financial support from MidAmerican Energy Company (MEC), and with input from scientists
at the U.S. Fish & Wildlife Service (USFWS) in support of development of a Habitat Conservation Plan for Wind Energy
Facilities in Iowa. The authors are grateful to Jesse Leckband, Senior Environmental Analyst at MEC for providing access to
the wind power facility and for procuring fresh bat carcasses for the experiments. We also thank Pablo Carrica, Ph.D.,
Research Engineer at IIHR-Hydroscience & Engineering for help in collecting the data. The funders had no role in the study
design, analysis of the data, decision to publish, or preparation of the manuscript.

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

455 0233/8/12/002.