# Peer review of "Experimental Investigation of Aerodynamic Characteristics of Bat Carcasses after Collision with a Wind Turbine"

_Wind Energy Science, 2019_

## Referee Comment (RC1) · Anonymous Referee #1 · 7 Aug 2019

General comments: Hundreds of thousands of bats are killed every year by spinning wind turbines. And the problem is increasing as the number of turbines increase worldwide. It is important to quantify the species and number of bats hit by the rotor blades to be able to evaluate the conservation aspects and potentially mitigate collisions. To determine the area that should be searched for dead bats under each turbine, this paper calculates the drag coefficient and develop ballistic models for real dead bats dropped from a high building and simultaneously surveyed by high-speed video. By making the first drag coefficient calculations of bats and implement them in models of fall distribution around turbines makes this paper useful and important in estimating the radius that must be searched. However, the experimental design was not optimal and

some uncertainty in estimating drag coefficients were found. The paper is well written and I have only minor comments.

Abstract and Conclusion: I would recommend to include the fall radius and how the results of this study should be implemented in bat conservation.

---

## Author Comment (AC1) · 4 Sep 2019

Dear Reviewer 1,

We thank the reviewer for a thoughtful review of our manuscript and providing valuable suggestions to highlight the impact of the research. We have provided a response to the comment below. The changes in response are added to the revised manuscript and have been marked in red for your convenience.

Best regards,

Shivendra Prakash and Corey D. Markfort

Response to Reviewer 1

Comment: Hundreds of thousands of bats are killed every year by spinning wind turbines. And the problem is increasing as the number of turbines increase world-wide. It is important to quantify the species and number of bats hit by the rotor blades to be able to evaluate the conservation aspects and potentially mitigate collisions. To determine the area that should be searched for dead bats under each turbine, this paper calculates the drag coefficient and develop ballistic models for real dead bats dropped a high building and simultaneously surveyed by high-speed video. By making the first drag coefficient calculations of bats and implement them in models of fall distribution around turbines makes this paper useful and important in estimating the radius that must be searched. However, the experimental design was not optimal and some uncertainty in estimating drag coefficients were found. The paper is well written and I have only minor comments.

Response: We thank the reviewer for commenting on the novelty of the research and impact it may have in addressing the challenge of determining the required fall radius around wind turbines to guide search efforts and carcass count estimation. Due to the limitations identified with obtaining empirical data on bat carcass aerodynamics, we acknowledge the main complication that falling carcasses rarely reach terminal velocity and therefore accelerations must be accounted for in the analysis. This is emphasized in the paper and a systematic analysis of the accelerating carcasses is provided.

Comment: Abstract and Conclusion: I would recommend to include the fall radius and how the radius of this study should be implemented in bat conservation.

Response: Thank you for the suggestion. We agree it is important to include details about how drag affects fall radius in both the Abstract and Conclusion section. We have appended the following text to the Abstract and Conclusion in the revised manuscript to emphasize the utility of obtaining the first estimates for bat carcass drag coefficients in order to make more robust estimations of fall radius, which impact implementation of

bat conservation.

Addition to the Abstract: The maximum range for bats falling after impact with a typical utility-scale onshore wind turbine was computed using the ballistic model. Based on the range of drag coefficient found in this research, Hoary and Evening bats are estimated to fall up to 109 m and 75 m, respectively. The ballistic model can be used to obtain fall distribution histograms for bats, employing the measured range of drag coefficient, to guide carcass survey efforts and correct survey data for limited or unsearched areas.

Additions to the Conclusion: For Hoary bat (heaviest bat) and Evening bat (lightest bat), the sensitivity of the bat carcass fall distribution generated from the ballistic model was investigated based on measured carcass mass and range of drag coefficient to determine the range of the maximum fall. Hoary bat, assuming the smallest $C_d$ (0.70), resulted in a maximum fall distance of 109 m, whereas Evening bat with largest $C_d$ (1.23) resulted in a maximum fall distance of 68 m.

The ballistic model framework proposed by HM10 generates 1D carcass fall zone histogram in the reference frame of the wind turbine rotor. The modelling framework can be extended by incorporating meteorological conditions such as wind speed and direction, resulting in a 2D fall zone histogram, representing the distribution of carcasses falling around the base of the turbine for a given period of time. Accounting for the distribution of wind direction, the model can be used to translate the histogram from the reference frame of the turbine to the reference frame of Earth. The resulting histogram can be compared to the carcass fall positions recorded in field surveys to validate the ballistic model, and guide search efforts. The model results may also be useful for correcting survey data for limited or unsearched areas, for example, when carcass surveys are conducted only on road and pads, or for a limited radius around turbines.

Please also note the supplement to this comment:
https://www.wind-energ-sci-discuss.net/wes-2019-33/wes-2019-33-AC1-
supplement.pdf

[Figure]

**Supplement:**

[revised manuscript text omitted]

---

## Referee Comment (RC2) · Jakob Mann (Referee) · 29 Feb 2020

The referee report has been uploaded as an Editor Comment.
* * *

---

## Editor Comment (EC1) · Jakob Mann (Editor) · 29 Feb 2020

First I have to apologise for the tardy review process on this paper. Many referees have been contacted without luck. Therefore will I, acting as an associate editor make a review myself.

This paper studies an area which is quite underrepresented in Wind Energy Science namely how to investigate bat killings by wind turbines. The specific research question investigated is how large a search area around the turbine is necessary to collect all the bat carcasses. This in turn depends on the aerodynamics of the carcasses and more specifically the terminal fall speed. The research is new and important but some

aspects have to be improved before it can be published. The paper is quite clear and well illustrated. Maybe some illustrations (Figs 9,10 , 12, 13) could be condensed or some parts left out.

Major issues 1. The so-called peak-locking error should be avoided. It is good but not sufficient to determine the top and bottom pixels for each frame of the bat carcass "shadow", but it is only the beginning of a more thorough analysis. It is well known from particle tracking studies that a much better resolution can be obtained (I guess that is also mentioned in the Westerweel reference). Already more than 20 years ago Mann, Ott and Andersen (1999) (Mann, J., Ott, S., & Andersen, J. S. (1999). Experimental study of relative, turbulent diffusion. Denmark. Forsknings-center Risoe. Risoe-R, No. 1036(EN) (page 22), see also Ott and Mann (2000), https://doi.org/10.1017/S0022112000001658 ) showed that precision down to 1/10 to 1/50 of a pixel is obtainable for images of dots that fill up a few pixels by a few pixels, very similar to the image in fig 6. The procedure is quite simple. First the background image (= no bats) should be subtracted from the image and then the already found top and bottom pixel values should be used to select a small area around the bat. Then the center of gravity of the pixel grey values should be calculated. It might not be that the precision will be as good as in the case of spherical particles but it should be much better still. 2. It is fine to calculate the drag coefficient Cd, but what is really important is the terminal fall velocity, wt. I think the emphasis should be on that because in the definition of Cd the projected area of the bat is entering. When this study is going to be used in practise people might be more interested in wt by species. There is quite some uncertainty related to the projected area. In table 1 there are some dimensions, but how exactly are those used to get the projected area? Are the wings of the bat flush with the body or do they sometimes flap out, greatly changing this area? These complications might be suppressed by focussing on the more practical wt.

Minor issues

1. The references in the start of the introduction are a bit outdated. See the recent re-

ports from either the International Renewable Energy Agency (IRENA) or WindEurope. You could also include other, newer references to wildlife issues related to wind energy, e.g. Kuik et al (2016) Wind Energy Science vol1 , pp 1.39 or H J Lindeboom et al (2011) Environmental Research Letters vol 6, 035101, or something else, just ass long as it is newer. 2. p 1 l 28. Could the wind turbine related bat fatality estimate 600.000 be related to the number of fatales caused other things, or maybe to the number of bats born every year. It might be difficult for the common reader of WES who has not much background knowledge in biology to relate to this number. Is it a lot or is it negligible? 3. also l 28: When you write a reference as a part of a sentence like Hayes 2013, "2013" has to be in parenthesis, see general WES guidlines on our web page. 4. p 2. It could be worthwhile to mention Sark and Sørensen (2015) Characterisation of blade throw from a 2.3 MW horizontal axis wind turbine upon failure, 53rd AIAA Aerospace Science meeting. There blade fragments can end up 100 to 500 m from the turbine. 5. p 3 l 71, Please mention the definition of coefficient of restitution e more clearly. 6. p 3 l 78, A "d" should be appended to "enable". 7. p 5 top lines. Use italics for the subscripts of the variables explained inline as they appear in the displayed equation (1). 8. p 5 eq 4, Minus inside cosh argument can be removed since cosh is symmetric. 9. p 7 table 1. It is unclear whether the numbers include wings or if they are only concerning the body. 10. p 8 text: Is it assumed in the analysis that the bat is falling on the y2 vertical line? Were the experiment conducted in no wind conditions? 11. p 11, l 210, How much was the lateral translation of the Evening bat? 12. p 11 sec 2.5: At the entrance of the bat into the frame of the camera, one could measure the position z and the vertical velocity w. The vertical velocity at that point could serve as the initial condition for (2) where wt is the only unknown. By adjusting wt the best fit to the subsequent measurements of w(t) could be obtained. I think I'm trying to simplify your procedure. 13. sec 2.5: it would be nice to avoid the complication of using Delta t_c. Maybe the procedure outlined in Major Issues point 1 would eventually make Delta t_c redundant. 14. p 19 l 358: Is the assumption of uniform distribution over the rotor right? 15. Section 5: It is not sufficiently well described how the histograms in figure 14 are obtained. What

is the assumption about the initial velocity of the bat relative to the blade speed at impact? The horizontal mean wind speed is an important parameters for the dispersion of the bats. What is assumed about that? How is it distributed? I think a more detail description is necessary. 16. Summary and Conclusion: It is not clear whether you use position or velocity data to do the fit. It is good to try to summarise the uncertainties, but the uncertainty on the fall distance should also be discussed.

―――――――――――――――――――

---

## Author Comment (AC2) · 19 Mar 2020

**Reply to Editor comments on "Experimental Investigation of Aerodynamic Characteristics of Bat Carcasses after Collision with a Wind Turbine" by Prakash and Markfort.**

Dear Dr. Mann,

Thank you for reviewing the manuscript and offering insightful suggestions and comments which helped improve the literature review, uncertainty in position values, and suggesting to include the terminal velocity estimates. We also thank the anonymous reviewer for valuable suggestions to highlight the impact of the research. We have provided a response to your comments below. The changes in the response are added to the revised manuscript and have been marked in red for your convenience. A supplemental information (SI) document is attached with the revised manuscript which contains the steps of the proposed multivariable optimization technique for drag coefficient estimation along with the Figs. 8, 10 and 12 from the original manuscript for the readers who might be interested in the in-depth details of the optimization process.

Sincerely,

Shivendra Prakash and Corey Markfort

**Response to Editor**

**Overview**

First, I have to apologize for the tardy review process on this paper. Many referees have been contacted without luck. Therefore will I, acting as an associate editor make a review myself.

This paper studies an area which is quite underrepresented in Wind Energy Science namely how to investigate bat killings by wind turbines. The specific research question investigated is how large a search area around the turbine is necessary to collect all the bat carcasses. This in turn depends on the aerodynamics of the carcasses and more specifically the terminal fall speed. The research is new and important, but some aspects have to be improved before it can be published. The paper is quite clear and well-illustrated. Maybe some illustrations (Figs 9, 10, 12, 13) could be condensed or some parts left out.

**Author response:** We thank the Editor for commenting on the novelty and importance of the present research to wind – wildlife interactions in Wind Energy Science which is currently underrepresented community. As suggested, we have condensed Figs. 9, 11, 13 in old manuscript in a single figure (Fig. 8 in revised manuscript). The systematic analysis of position extraction from high – speed images, species terminal velocity and response to the minor comments is presented below.

**Editor comment:**

**Major issue 1.**

The so-called peak-locking error should be avoided. It is good but not sufficient to determine the top and bottom pixels for each frame of the bat carcass "shadow", but it is only the beginning of a more thorough analysis. It is well known from particle tracking studies that a much better resolution can be obtained (I guess that is also mentioned in the Westerweel reference). Already more than 20 years ago Mann, Ott and Andersen (1999) (Mann, J., Ott, S., & Andersen, J. S. (1999). Experimental study of relative, turbulent diffusion. Denmark. Forskningscenter Risoe. Risoe-R, No. 1036(EN) (page 22), see also Ott and Mann (2000), https://doi.org/10.1017/S0022112000001658) showed that precision down to 1/10 to 1/50 of a pixel is obtainable for images of dots that fill up a few pixels by a few pixels, very similar to the image in fig 6. The procedure is quite simple. First the background image (= no bats) should be subtracted from the image and then the already found top and bottom pixel values should be used to select a small area around the bat. Then the center of gravity of the pixel grey values should be calculated. It might not be that the precision will be as good as in the case of spherical particles, but it should be much better still.

**Authors response:** Thank you for the suggestion and for pointing out the useful references and methods for improving the high – speed image analysis for the carcass drop experiments. The discussion on peak – locking error has been removed. The proposed methodology determines the position on images with the precision of 0.10 pixels – 0.02 pixels, by fitting a Gaussian function to the particle pixel grey intensity.

We show here results from the recommended procedure as applied to the position measurements of drop #1 of Hoary bat carcass experiment. The carcass position measurements from the recommended method is compared with the earlier measurements obtained from carcass top and bottom pixel coordinates. Figure R1 shows the snapshot of the greyscale image at $t = 0.420$ s during drop #1 of Hoary bat carcass. The image has been cropped to focus on the region around the carcass image. The greyscale intensity between $0 – 255$ is described at each pixel coordinate encompassed in Fig. R1. The rectangular area enclosing the carcass and surrounding interrogation region covers 15 pixels (horizontally) and 13 pixels (vertically). The carcass image occupies eight pixels or 57 mm (1 pixel = 7.10 mm). The coordinates of each pixel in horizontal ($x_{pixel}$) and vertical ($z_{pixel}$) direction and pixel intensities ($I_{pixel}$) displayed in Fig. R1 were recoded. The pixel intensities were summed over all $x_{pixels}$ to yield the total pixel intensity ($I_{Total}$) as a function of $z_{pixel}$. Table T1 shows $I_{Total}$ at each $z_{pixel}$ of the rectangular area in Fig. R1.

[Figure]

**Fig. R1: Selected grey area pixels around carcass at *t* = 0.420 s**

**Table T1: $I_{Total}$ at a given $z_{pixel}$**

| $z_{pixel}$ | $I_{Total}$ |
|---|---|
| 155 | 34 |
| 156 | 65 |
| 157 | 165 |
| 158 | 788 |
| 159 | 1383 |
| 160 | 1503 |
| 161 | 1501 |
| 162 | 1378 |
| 163 | 1251 |
| 164 | 1020 |
| 165 | 413 |
| 166 | 46 |
| 167 | 30 |

Figure R2 shows $z_{pixel}$ vs. $I_{Total}$ plot (red dots) for the measurements in Table T1. The centroid of greyscale intensity in Table T1 was computed by averaging each $z_{pixel}$ based on its $I_{Total}$ value (Eq. (E1)) and was found to be 161.123. The carcass centroid from top and bottom pixel coordinate was estimated as 160. This procedure was repeated for next four consecutive images and carcass centroid was calculated for each of the image frames.

$$z_{c,new} = \frac{\sum I_{pixel} z_{pixel}}{\sum I_{pixel}}$$

(E1)

Table T2 below displays the comparison of $z_c$ from two methods, i.e., Mean top/bottom method (2nd column) and center of intensity method (3rd column) for the selected greyscale frames. It is evident from the table that maximum $z_c$ difference is of the order of approximately 1 pixel from the two methods.

**Table T2: Comparison of $z_c$**

| $t$ (s) | $z_c$ (Original mean top/bottom method) (m) | $z_c$ (Center of Intensity method, E1) (m) | $|\Delta z_c|$ (mm) |
|---------|---------------------------------------------|--------------------------------------------|---------------------|
| 0.420 | 5.928 | 5.919 | 9 |
| 0.424 | 5.903 | 5.900 | 3 |
| 0.428 | 5.879 | 5.876 | 3 |
| 0.432 | 5.857 | 5.853 | 4 |
| 0.436 | 5.836 | 5.830 | 6 |

Mann et al. (1999) proposed fitting particle images with a Gaussian shape function to determine the position from the images. A similar procedure was applied to the measured distribution shown in Fig. R2 to compute the centroid of the distribution, $z_c$. The Gaussian distribution expressed by Eq. (E2) was fitted to $z_{pixel}$ vs. $I_{Total}$ data in Table T1 to find the best estimate of carcass centroid, $z_c$, at $t = 0.420$ s. Figure R2 shows the Gaussian distribution (black line) fitted to $z_{pixel}$ vs. $I_{Total}$ measurements to give $z_c = 160.988$.

$$I_{Total} = \frac{1}{\sigma_z\sqrt{2\pi}}e^{-\frac{1}{2}\left(\frac{z_{pixel}-z_c}{\sigma_z}\right)^2} \tag{E2}$$

[Figure]

**Fig. R2: Gaussian distribution fitted to $z_{pixel}$ vs. $I_{Total}$ measurements**

The Gaussian distribution was fitted to $z_{pixel}$ vs. $I_{Total}$ measurements of the images to determine the respective centroid. Table T3 shows the comparison of the carcass centroid estimated using top and bottom pixels (2nd column) and by fitting Gaussian distribution (3rd column) to $z_{pixel}$ vs. $I_{Total}$ measurements. Again, we notice from Table T3 that the difference in the $z_c$ estimates from the two approaches are of the order of 1 pixel.

**Table T3: Comparison of $z_c$**

| $t$ (s) | $z_c$ (Top and bottom pixel) (m) | $z_c$ (Gaussian fitting) (m) | $|\Delta z_c|$ (mm) |
|---------|---------|---------|---------|
| 0.420 | 5.928 | 5.920 | 8 |
| 0.424 | 5.903 | 5.898 | 5 |
| 0.428 | 5.879 | 5.875 | 4 |
| 0.432 | 5.857 | 5.852 | 5 |
| 0.436 | 5.836 | 5.829 | 7 |

The procedure of fitting a Gaussian distribution to $z_{pixel}$ vs. $I_{Total}$ measurements to determine, $t$ vs. $z_c$, was applied to Hoary bat drop #1 experiment images corresponding to the time instants shown in Fig. 9 of the old manuscript (Fig. 8 in the revised manuscript). The objective was to determine how the difference in position estimates affect the initial drop height ($z_0$), carcass drag coefficient ($C_d$) and terminal velocity ($w_t$). Table T4 below presents the comparison of centroid ($z_c$) of the Hoary bat carcass computed from carcass top and bottom pixel coordinates (column 2) and by fitting a Gaussian function (column 3) to $z_{pixel}$ vs. $I_{Total}$ measurements, respectively. Column 4 shows the maximum difference of approximately 1.5 pixels in the $z_c$ estimates from two methods.

**Table T4: Comparison of $z_c$ values from two methods**

| $t$ (s) | $z_c$ (Original mean top/bottom method) (m) | $z_c$ (m), (Gaussian fit method, E2) (m) | difference $|\Delta z_c|$ (mm) |
|---------|---------|---------|---------|
| 0.591 | 5.904 | 5.898 | 6 |
| 0.695 | 5.264 | 5.259 | 5 |
| 0.799 | 4.547 | 4.547 | 0 |
| 0.903 | 3.752 | 3.744 | 8 |
| 1.007 | 2.857 | 2.852 | 5 |
| 1.111 | 1.896 | 1.885 | 11 |
| 1.215 | 0.845 | 0.841 | 4 |

The carcass positions computed with the recommended method of Gaussian fit (Table T4) were used to find the carcass velocity. The ballistics model was then fitted to the velocity measurements to determine the optimal drop height ($z_0$), carcass drag coefficient ($C_d$) and subsequently terminal velocity ($w_t$). Table T5 shows the comparison of $z_0$, $C_d$, $w_t$ and corresponding percent change, computed using the positions from the two methods. It is evident from the table that the $C_d$ and $w_t$ computed by fitting the ballistics model to the velocity measurements of Gaussian fit method is same as the old computations of mean top/bottom method. However, there is a small difference of 0.26 % in optimized $z_0$ computations.

Figure R3 shows the comparison of the analytical solution of carcass position and velocity of the 1-D ballistics model, with the experimental measurements from two methods to test the accuracy of the new $C_d$ estimates. The comparison shows that experimental measurements from both methods are in good agreement with the analytical solution for both position and velocity.

Both the average centre of the carcass image, and the centre of the intensity distribution, represent approximations of the centre of gravity. The original approach of top/bottom method assumes uniform weight distribution whereas the centre of intensity approach assumes the weight distribution to be related with the greyscale intensity distribution, which may not necessarily be the case.

**Table T5: Comparison of carcass drag coefficient, terminal velocity and percent change from two methods**

|  | Mean top/bottom method | Gaussian fit method | % difference |
|---|---|---|---|
| $C_d$ | 0.70 | 0.70 | 0 |
| $w_t$ (m/s) | 17.57 | 17.57 | 0 |
| $z_0$ (m) | 7.58 | 7.56 | 0.26 |

[Figure]

**Fig. R3: Comparison of position and velocity from new $C_d$ estimate (analytical model vs. measured data)**

**Major issue 2.**

It is fine to calculate the drag coefficient $C_d$, but what is really important is the terminal fall velocity, $w_t$. I think the emphasis should be on that because in the definition of $C_d$ the projected area of the bat is entering. When this study is going to be used in practice people might be more interested in $w_t$ by species. There is quite some uncertainty related to the projected area. In table 1 there are some dimensions, but how exactly are those used to get the projected area? Are the wings of the bat flush with the body or do they sometimes flap out, greatly changing this area? These complications might be suppressed by focusing on the more practical $w_t$.

**Authors response:** We agree it is important to report the terminal velocity and indicate its practical importance to the Wind Energy Science community. Haider and Levenspiel (1989) also mentioned that, it is the terminal velocity, rather than the drag coefficient, that is of interest. However, when either of drag coefficient or terminal velocity of the carcass is known, the other can be computed using the mass and projected area features, by equating the drag force equal to the gravitational force. In the present study, the carcass terminal velocity was determined for each of the species based on the computed drag coefficients and is reported in Table 3 and Table 5 of the

revised manuscript. Additional details describing the projected area estimation procedure is added in the revised manuscript (Page #8, line #191 – 205).

"Firstly, the irregularly shaped carcass was approximated as an ellipsoid (Fig. 3), where $a$ and $b$ = c represents the dimensions along the ellipsoid major and both minor axes, respectively. The carcass projected area ($A_p$) was computed as $\Pi/4\ d_{eq}^2$, where $d_{eq}$ is the equivalent diameter of the sphere having the same volume as that of the ellipsoid shaped bat. The equivalent diameter (Eq. (6)) is determined by equating the volume of a sphere (left hand side) with an ellipsoid (right hand side) in Eq. (5).

$$\frac{\pi}{6}d_{eq}^3 = \frac{4}{3}\pi abc \tag{5}$$

$$d_{eq} = 2\sqrt[3]{abc} \tag{6}$$

The wings of the bat are assumed to be folded and flush with the body during the fall as observed from the high – speed images of the carcass drop experiments, which as a result eliminates a source of uncertainty in the projected area computation. It is possible that the wing may be severely broken during blade strike, and therefore this simplification may not be applicable in all carcass fall cases."

**Minor issues**

1.  The references in the start of the introduction are a bit outdated. See the recent reports from either the International Renewable Energy Agency (IRENA) or WindEurope. You could also include other, newer references to wildlife issues related to wind energy, e.g. Kuik et al (2016) Wind Energy Science vol1 , pp 1.39 or H J Lindeboom et al (2011) Environmental Research Letters vol 6, 035101, or something else, just as long as it is newer.

**Authors response:** Thank you for suggesting newer references related to wind – wildlife interactions. These references have been included in the revised manuscript.

2.  p 1 | 28. Could the wind turbine related bat fatality estimate 600,000 be related to the number of fatales caused other things, or maybe to the number of bats born every year. It might be difficult for the common reader of WES who has not much background knowledge in biology to relate to this number. Is it a lot or is it negligible?

**Authors response:** We have addressed this question in the revised manuscript (Page #2, line #43 – 50)

"Several studies have applied different methods to estimate the number of bats killed at wind energy facilities in the United States. Kunz et al. (2007) used model and survey data to estimate bat fatalities and found that annually 33,000 – 111,000 bats would be killed at wind energy fatalities. Cryan (2011) estimated annual bat fatalities at 450,000 is North America based on a bat fatality rate of 11.60 bats/MW/yr. While Smallwood (2013) estimated 888,000 bats were killed in the United States in 2012; Hayes (2013) concluded that, in 2012, over 600,000 bats

are likely to have died in the United States as a result of direct interactions with wind turbines. Since many bat species give birth to only one pup per year and bats have high mortality rates during the first year of their life (O'shea et al. (2004), Hallam and Federico (2009)), the estimated numbers of annual bat fatalities is considered to be significant."

3.  also | 28: When you write a reference as a part of a sentence like Hayes 2013, "2013" has to be in parenthesis, see general WES guidelines on our web page.

**Authors response:** We corrected the revised manuscript.

4.  p 2. It could be worthwhile to mention Sark and Sørensen (2015). Characterization of blade throw from a 2.3 MW horizontal axis wind turbine upon failure, 53rd AIAA Aerospace Science meeting. There blade fragments can end up 100 to 500 m from the turbine.

**Authors response:** We added the reference in the revised manuscript. (Page # 3, line #80 – 87)

5.  p 3 | 71, Please mention the definition of coefficient of restitution $e$ more clearly.

**Authors response:** We have added this in the revised manuscript. (Page # 3, line #98 – 102)

6.  p 3 | 78, A "$d$" should be appended to "enable".

**Authors response:** Corrected in the revised manuscript.

7.  p 5 top lines. Use italics for the subscripts of the variables explained inline as they appear in the displayed equation (1).

**Authors response:** Corrected in the revised manuscript.

8.  p 5 eq 4, Minus inside cosh argument can be removed since cosh is symmetric.

**Authors response:** Implemented in the revised manuscript.

9.  p 7 table 1. It is unclear whether the numbers include wings or if they are only concerning the body.

**Authors response:** The carcass dimensions reported in Table 1 are based on the body only as the analysis of the high – speed images of carcass drop experiment revealed that bat wings were mostly flush with the body during the fall.

10. p 8 text: Is it assumed in the analysis that the bat is falling on the $y_2$ vertical line? Were the experiment conducted in no wind conditions?

**Authors response:** Freshly collected carcasses were dropped in front of a 6.30 m high wall on the leeward side of a building to achieve approximately quiescent or no wind conditions. As seen in Fig. 4 (revised manuscript), the analysis of the high – speed video of the carcass drop experiments showed that carcasses movement horizontally is negligible except for the Evening Bat carcass, which experienced oscillating lateral translation with an amplitude on the order of 10 cm. Based on our observations, carcasses can be assumed to fall along the vertical line at distance $y_2$ from the wall.

11. p 11 | 210: How much was the lateral translation of the Evening bat?

**Authors response:** The lateral displacement of the Evening Bat between when the carcass enters the field of view at a height of 5.87 m above the ground and when the carcass reaches the ground was measured to be 10 cm. Therefore, the lateral displacement of the Evening bat was found to be 1.80% of the carcass drop distance and was assumed to be negligible for estimating the trajectories.

12. p 11 sec 2.5: At the entrance of the bat into the frame of the camera, one could measure the position $z$ and the vertical velocity $w$. The vertical velocity at that point could serve as the initial condition for (2) where $w_t$ is the only unknown. By adjusting $w_t$ the best fit to the subsequent measurements of $w$ ($t$) could be obtained. I think I'm trying to simplify your procedure.

**Authors response:** Thank you for suggesting a method for simplifying the optimization procedure. However, the key issue here is that the falling carcass is accelerating in phase II (Fig. 2 in the revised manuscript) of the fall curve. Therefore, the initial velocity obtained from the central – difference scheme on position time series even at the top of the field of view of the image depends on the choice of $\Delta t_c$. A short time separation can cause the velocity difference in the same range as the precision error for the velocity, leading to velocity material derivative affected by large random error. On the other hand, the truncation error in central – differencing increases with large time separation window. The optimal choice of time separation to compute velocity is achieved via a trade – off between random and bias errors (Novara and Scarano (2013)). Without $\Delta t_c$, the velocity at a time instant cannot be computed and subsequently cannot be used as an initial velocity ($w_0$) to determine $w_t$ from Eq. (2). Our approach provides a method to optimize for initial conditions ($z_0$, $w_0$), optimal time filtering window ($\Delta t_c$) and drag coefficient ($C_d$) to obtain the best comparison between the measured velocity and fall model velocity.

13. sec 2.5: it would be nice to avoid the complication of using $\Delta t_c$. Maybe the procedure outlined in Major Issues point 1 would eventually make $\Delta t_c$ redundant.

**Authors response:** The high – speed imaging of the carcass drop experiments provides the opportunity to track the same accelerating carcass over multiple images. These images provides the correlated velocity or acceleration measurements at different locations along the carcass trajectory. The drawback of the particle imaging techniques is that they rely on measuring particle displacement. The displacement spatial and temporal derivatives are inevitably subjected to noise amplification if the issue of optimal time window ($\Delta t_c$) for computing derivatives is not resolved (Toshi and Sega (2019), Chapter 6). The response to issue #11 above highlights the challenges in identifying the appropriate filtering window ($\Delta t_c$).

Westerweel (1997) showed that instead of pixel size, the particle – image diameter relative to the size of interrogation domain determines the measurement resolution. The information with regard to particle – image displacement does not improve with particle image having diameter more than two pixels. Westerweel (1997) also mentioned that if there is velocity gradient over the interrogation domain, then the number of measurements from the smaller displacements are larger than that of the larger displacements, known as velocity bias. As a result, the measured displacement is biased towards the smaller displacements. However, the velocity bias also occurs from uniform displacements.

Keeping all the above – mentioned challenges in mind, the proposed multivariable optimization method suggests one of the ways to find $\Delta t_c$ using which the robust fitting can be performed to compute $z_0$ and $C_d$.

14.  p 19 | 358: Is the assumption of uniform distribution over the rotor right?

**Authors response:** The purpose of the sensitivity analysis in section 5 of the manuscript was to highlight the importance of the relative effect of carcass mass and drag coefficient on the carcass fall zone histogram for a given distribution of bats on the rotor, meteorological features and corresponding turbine operational conditions. However, there is no literature on the exact strike distribution of the bats on the rotor plane. Therefore, as a first approximation, bats were assumed to be distributed on the rotor plane at a radial resolution of 1 m and angular resolution of 5°, resulting in approximately 4000 bat carcasses striking the rotor.

15.  Section 5: It is not sufficiently well described how the histograms in figure 14 are obtained. What is the assumption about the initial velocity of the bat relative to the blade speed at impact? The horizontal mean wind speed is an important parameter for the dispersion of the bats. What is assumed about that? How is it distributed? I think a more detail description is necessary.

**Authors response:** We agree a more thorough description of the model generating Fig. 14 (Fig. 9 in the revised manuscript) is needed. We have added details of the 2-D ballistics model in the revised manuscript (Page #17, line #346 – 355).

"The sensitivity of bat fall zone distributions in the rotor plane (following the modelling approach of HM10) with respect to carcass mass and its drag coefficient was tested. Hoary and Evening bat were selected for this exercise because they are the heaviest and lightest bats, respectively. Figure 9 shows fall zone distributions for Hoary bat (upper row) and Evening bat (lower row) for the highest and lowest values of $C_d$, respectively. The distributions were obtained by solving the 2-D ballistics trajectories in quiescent flow for bats assumed to be impacted by turbine blades within the rotor plane. However, there is no literature on the exact strike distribution of the bats on the rotor plane. Therefore, as a first approximation, bats were assumed to be distributed on the rotor plane at a radial resolution of 1 m and angular resolution of 5°, resulting in approximately 4000 bat carcasses striking the rotor. The coefficient of restitution ($e$) was assumed as zero during the computations which means bat carcass attains the same velocity as the rotor at the point of impact, i.e., at impact the initial velocity of bat relative to the blade is zero. In these simulations, the rotor radius is 54 m, hub height is 80 m and the turbine RPM is 8.70."

We agree that horizontal mean wind speed is an important parameter for the dispersion of the bats. We are currently investigating the effect of background wind on carcass trajectories, and so far have found that wind affects the shape of the histogram and while the maximum distance is similar for Hoary bat, it is significantly larger for Evening bat due to the effect of wind drift.

16. Summary and Conclusion: It is not clear whether you use position or velocity data to do the fit. It is good to try to summarize the uncertainties, but the uncertainty on the fall distance should also be discussed.

**Authors response:** The velocity data was used to fit the ballistics model (described in multivariable optimization algorithm steps presented now in SI) to determine the carcass drag coefficient. The discussion on fall distance uncertainties is added in the revised manuscript (Page #19, line #379– 384).

"The range of the maximum fall zone for the Hoary bat (heaviest) and Evening bat (lightest) were investigated with a ballistics model, and the sensitivity of the bat carcass fall zone distributions based on the measured carcass mass and range of drag coefficient were determined. Hoary bat, assuming the smallest $C_d$ = 0.70, resulted in a maximum fall distance of $X_{max}$ = 92 m, whereas Evening bat with largest $C_d$ = 1.23 resulted in a maximum fall distance of $X_{max}$ = 58 m. This demonstrates the relative effect of bat mass and carcass aerodynamics have significant influence on maximum distance travelled by bats after strike by a turbine blade."

We are examining the effect of horizontal mean wind speed on carcass fall trajectories and discovered that the maximum distance is similar for Hoary bat but significantly larger for Evening bat due to the wind drift effect.

**References:**

Mann, J., Ott, S. and Anderson, J. S. (1999). Experimental study of relative, turbulent diffusion, Denmark. Forskningscenter Risoe. Risoe – R, No. 1036 (EN).

Novara, M. and Scarano, F. (2013). A particle – tracking approach for accurate material derivative measurements with tomographic PIV. Exp Fluids, 54:1584. DOI 10.1007/s00348-013-1584-5.

Ott, S. and Mann, J. (2000). An experimental investigation of the relative diffusion of particle pairs in three – dimensional turbulent flow. J. Fluid Mech., vol. 422, pp. 207 – 223. DOI: https://doi.org/10.1017/S0022112000001658.

Toshi, F. and Sega, M (2019). Flowing matter. Springer Publishing House.

Westerweel, J. (1997). Fundamentals of digital particle image velocimetry. Meas. Sci. Technol., 8, 1379-92. 10.1088/0957-0233/8/12/002.

---

## Author Comment (AC3) · 19 Mar 2020

[revised manuscript text omitted]